# SEMANTIC HIERARCHY EMERGES IN DEEP GENERATIVE REPRESENTATIONS FOR SCENE SYNTHESIS

## ABSTRACT

Despite the success of Generative Adversarial Networks (GANs) in image synthesis, there lacks enough understanding on what networks have learned inside the deep generative representations and how photo-realistic images are able to be composed from random noises. In this work, we show that highly-structured semantic hierarchy emerges as variation factors for synthesizing scenes from the generative representations in state-of-the-art GAN models, like StyleGAN and BigGAN. By probing the layer-wise representations with a broad set of semantics at different abstraction levels, we are able to *quantify* the causality between the activations and semantics occurring in the output image. Such a quantification identifies the human-understandable variation factors learned by GANs to compose scenes. The qualitative and quantitative results suggest that the generative representations learned by the GANs with layer-wise latent codes are specialized to synthesize different hierarchical semantics: the early layers tend to determine the spatial layout and configuration, the middle layers control the categorical objects, and the later layers finally render the scene attributes as well as color scheme. Identifying such a set of manipulatable latent variation factors facilitates semantic scene manipulation[1].

## 1 INTRODUCTION

Success of deep neural networks stems from the representation learning, which identifies the explanatory factors underlying the high-dimensional observed data (Bengio et al. (2013)). Prior work has shown that many concept detectors spontaneously emerge inside the deep representations trained for classification task. For example, Gonzalez-Garcia et al. (2018) shows that networks for object recognition are able to detect semantic object parts, and Bau et al. (2017) confirms that deep representations from classifying images learn to detect different categorical concepts at different layers.

Analyzing the deep representations and their emergent structures gives insight into the generalization ability of deep features (Morcos et al. (2018)) as well as the feature transferability across different tasks (Yosinski et al. (2014)). But current efforts on interpreting deep representations mainly focus on discriminative models (Zhou et al. (2015); Gonzalez-Garcia et al. (2018); Zeiler and Fergus (2014); Agrawal et al. (2014); Bau et al. (2017)). Recent advance of Generative Adversarial Networks (GANs) (Goodfellow et al. (2014); Karras et al. (2018a;b); Brock et al. (2019)) is capable of transforming random noises into high-quality images, however, the nature of the learned generative representations and how a photo-realistic image is being composed over different layers of the generator in GAN remain much less explored.

It is known that the internal units of Convolutional Neural Networks (CNNs) emerge as object detectors when trained to categorize scenes (Zhou et al. (2015)). Representing and detecting informative categorical objects provides an ideal solution for classifying scenes, such as sofa and TV are representative of living room while bed and lamp are of bedroom. However, synthesizing a scene demands far more knowledge for the generative models to learn. Specifically, in order to produce highly-diverse scene images, the deep representations might be required to not only generate every individual object relevant to a specific scene category, but also decide the underlying room layout as well as render various scene attributes, *e.g.*, the lighting condition and color scheme. Very recent work

---

[1]Source code will be made available. Please see the demo video at this link.

on interpreting GANs Bau et al. (2019) visualized that the internal filters at intermediate layers are specialized for generating some certain objects, but studying scene synthesis from object aspect only is far from fully understanding how GAN is able to compose a photo-realistic image, which contains multiple variation factors from layout level, category level, to attribute level. The original StyleGAN work (Karras et al. (2018b)) pointed out that the layer-wise latent codes actually control the synthesis from coarse to fine, but how these variation factors are composed together and how to quantify such semantic information are still uncertain. Differently, this work gives a much deeper interpretation on the hierarchical generative representations in the sense that we match these layer-wise variation factors with human-understandable scene variations at multiple abstraction levels, including *layout*, *category (object)*, *attribute*, and *color scheme*.

Starting with the state-of-the-art StyleGAN models (Karras et al. (2018b)) as the example, we reveal that highly-structured semantic hierarchy emerges from the deep generative representations with layer-wise stochasticity trained for synthesizing scenes, even without any external supervision. Layer-wise representations are first probed with a broad set of visual concepts at different abstraction levels. By quantifying the causality between the layer-wise activations and the semantics occurring in the output image, we are able to identify the most relevant variation factors across different layers of a GAN model with layer-wise latent codes: the early layers specify the spatial layout, the middle layers compose the category-guided objects, and the later layers render the attributes and color scheme of the entire scene. We further show that identifying such a set of manipulatable latent variation factors from layouts, objects, to scene attributes and color schemes facilitates the semantic image manipulation with large diversity. The proposed manipulation technique is further generalized to other GANs such as BigGAN (Brock et al. (2019)) and ProgressiveGAN (Karras et al. (2018a)).

## 1.1 RELATED WORK

**Deep representations from classifying images.** Many attempts have been made to study the internal representations of CNNs trained for classification tasks. Zhou et al. (2015) analyzed hidden units by simplifying the input image to see which context region gives the highest response, Simonyan et al. (2014) applied back-propagation technique to compute the image-specific class saliency map, Bau et al. (2017) interpreted the hidden representations via the aid of segmentation mask, Alain and Bengio (2016) trained independent linear probes to analyze the information separability among different layers. There are also some studies transferring the features of CNNs to verify how learned representations fit with different datasets or tasks (Yosinski et al. (2014); Agrawal et al. (2014)). In addition, reversing the feature extraction process by mapping a given representation back to image space (Zeiler and Fergus (2014); Nguyen et al. (2016); Mahendran and Vedaldi (2015)) also gives insight into what CNNs actually learn to distinguish different categories. However, these interpretation techniques developed for classification networks cannot be directly applied for generative models.

**Deep representations from synthesizing images.** Generative Adversarial Networks (GANs) (Goodfellow et al. (2014)) advance the image synthesis significantly. Some recent models (Karras et al. (2018a); Brock et al. (2019); Karras et al. (2018b)) are able to generate photo-realistic faces, objects, and scenes, making GANs applicable to real-world image editing tasks, such as image manipulation (Shen et al. (2018); Xiao et al. (2018a); Wang et al. (2018); Yao et al. (2018)), image painting (Bau et al. (2019); Park et al. (2019)), and image style transfer (Zhu et al. (2017); Choi et al. (2018)). Despite such a great success, it remains uncertain what GANs have actually learned to produce such diverse and realistic images. Radford et al. (2016) pointed out the vector arithmetic phenomenon in the underlying latent space of GAN, however, discovering what kinds of semantics exist inside a well-trained model and how these semantics are structured to compose high-quality images are still unsolved. A very recent work (Bau et al. (2019)) analyzed the individual units of the generator in GAN and found that they learn to synthesize informative visual contents such as objects and textures spontaneously. Unlike Bau et al. (2019) which focuses on the intermediate filters, our work quantitatively explores the emergence of multi-level semantics inside the very early latent space.

## 2 VARIATION FACTORS FOR SCENE SYNTHESIS

### 2.1 MULTI-LEVEL SCENE SEMANTICS

Imagine an artist drawing a picture of living room. The very first step, before drawing every single object, is to choose a perspective and set up the layout of the room. After the spatial structure is

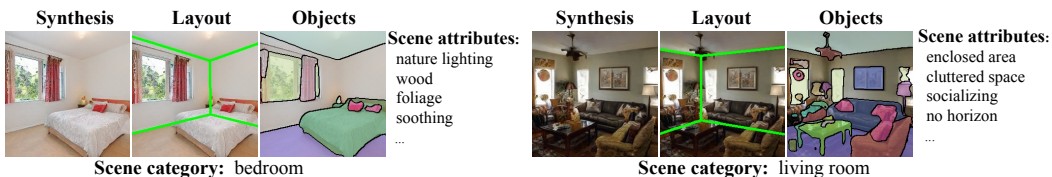

Figure 1: Multiple levels of semantics extracted from two synthesized scenes.

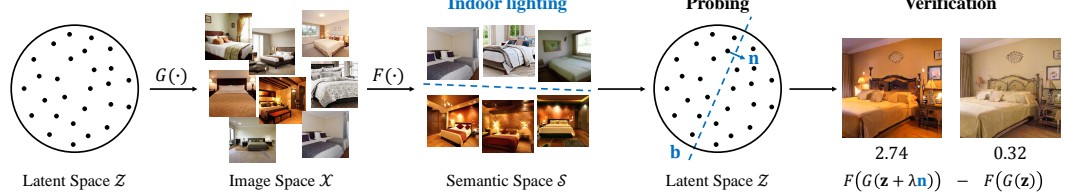

Figure 2: Method for identifying the emergent variation factors in generative representation. By deploying a broad set of *off-the-shelf* image classifiers as scoring functions, $F(\cdot)$, we are able to assign a synthesized image with semantic scores corresponding to each candidate variation factor. For a particular concept, we learn a decision boundary in the latent space by considering it as a binary classification task. Then we move the sampled latent code towards the boundary to see how the semantic varies in the synthesis, and use a re-scoring technique to quantitatively verify the emergence of the target concept.

decided, the next step is to add objects that typically occur in a living room, such as sofa and TV. Finally, the artist will refine the details of the picture with specified decoration styles, *e.g.*, warm or cold, natural lighting or indoor lighting. The above process reflects how a human interprets a scene to draw it. As a comparison, generative models such as GANs follow a completely end-to-end training for synthesizing scenes, without any prior knowledge about the drawing techniques and relevant concepts. Even so, the trained GANs are able to produce photo-realistic scenes, which makes us wonder if the GANs have mastered any human-understandable drawing knowledge as well as the variation factors of scenes spontaneously.

Therefore, in this work we aim at interpreting how GANs learn to synthesize a photo-realistic scene image from scratch. To align the synthesized scenes with human perception, we use off-the-shelf classifiers to extract semantics from the output image. As shown in Fig.1, given a scene image, semantics at multiple abstraction levels are extracted, including layout, object (category), and attribute. These concepts are treated as candidates and we propose a quantification technique in Sec.2.2 to identify which variation factor has been encoded into the well-learned generative representation. We surprisingly find that GAN synthesizes a scene in a manner highly consistent with human. Over the convolutional layers, GAN manages to compose these multi-level abstractions hierarchically. In particular, GAN constructs the spatial layout at early stage, synthesizes category-specified objects at middle stage, and renders the scene attribute (*e.g.*, color scheme) at later stage. We will describe the method we use to quantify the emergent variation factors as follows.

## 2.2 Identifying the Emergent Variation Factors

Among the multi-level candidate concepts described in Sec.2.1, not all of them are meaningful to a particular scene synthesis model. For instance, "indoor lighting" will never happen in outdoor scenes such as bridge and tower. Accordingly, we come up with a method to quantitatively identify the variation factors that emerge inside the learned generative representation. Fig.2 illustrates the identification process which consists of two steps, *i.e.*, probing and verification.

**Probing latent space.** The generator of GAN, $G(\cdot)$, typically learns the mapping from latent space $\mathcal{Z}$ to image space $\mathcal{X}$. Latent vectors $\mathbf{z} \in \mathcal{Z}$ can be considered as the generative representation learned by GAN. To study the emergence of variation factors inside $\mathcal{Z}$, we need to first extract semantic information from $\mathbf{z}$, which is not trivial. To solve this problem, we employ synthesized image, $\mathbf{x} = G(\mathbf{z})$, as an intermediate step and use a broad set of *off-the-shelf* image classifiers to help assign semantic scores for each sampled latent code $\mathbf{z}$. Taking "indoor lighting" as an example, the scene attribute classifier is able to output the probability on how an input image looks like having indoor lighting, which we use as semantic score. Recall that we divide scene representation into layout, category, and attribute levels, we introduce layout estimator, scene category recognizer, and attribute classifier to predict semantic scores from these abstraction levels respectively, forming a hierarchical

semantic space $\mathcal{S}$. After establishing the one-on-one mapping from latent space $\mathcal{Z}$ to sematic space $\mathcal{S}$, we search the decision boundary for each concept by treating it as a bi-classification problem, as shown in Fig.2. Here, taking "indoor lighting" as an instance, the boundary separates the latent space $\mathcal{Z}$ to two sets, *i.e.*, present or absent of indoor lighting.

**Verifying manipulatable variation factors.** After probing the latent space with a broad set of candidate concepts, we still need to figure out which ones are most relevant to the generative model acting as the variation factors. The key issue is how to define "relevance", or say, how to verify whether the learned representation has already encoded a particular variation factor. We argue that if the target concept is manipulatable from latent space perspective (*e.g.*, change the indoor lighting status of the synthesized image via simply varying the latent code), the GAN model is able to capture such variation factors during the training process.

As mentioned above, we have already got separation boundaries for each candidate. Let $\{\mathbf{n}_i\}_{i=1}^C$ denote the normal vectors of these boundaries, where $C$ is the total number of candidates. For a certain boundary, if we move a latent code $\mathbf{z}$ along its normal direction (positive), the semantic score should also increase correspondingly. Therefore, we propose to re-score the varied latent code to *quantify* how a variation factor is relevant to the target model for analysis. As shown in Fig.2, this process can be formulated as

$$\Delta s_i = \frac{1}{K} \sum_{k=1}^K \max \Big( F_i\big(G(\mathbf{z}^k + \lambda \mathbf{n}_i)\big) - F_i\big(G(\mathbf{z}^k)\big), 0 \Big), \tag{1}$$

where $\frac{1}{K} \sum_{k=1}^K$ stands for the average of $K$ samples to make the metric more accurate. $\lambda$ is a fixed moving step. To make this metric comparable among all candidates, all normal vectors $\{\mathbf{n}_i\}_{i=1}^C$ are normalized to fixed norm 1 and $\lambda$ is set as 2. With this re-scoring technique, we can easily rank the score $\Delta s_i$ among all $C$ concepts to retrieve the most relevant latent variation factors.

## 3 EXPERIMENTAL RESULTS

In the generation process, the deep representation at each layer (especially for StyleGAN and BigGAN) is actually directly derived from the projected latent code. Therefore, we consider the latent code as the "generative representation", which may be slightly different from the conventional definition in the classification networks. We conduct a detailed empirical analysis on the variation factors identified across the layers of the generators in GANs. We show that the hierarchy of variation factors emerges in the deep generative representations as a result of synthesizing scenes. Sec.3.1 contains the layer-wise analysis on the state-of-the-art StyleGAN model (Karras et al. (2018b)), quantitatively and qualitatively verifying that the multi-level variation factors are encoded in the latent space. In Sec.3.2 we explore the question on how GANs represent categorical information such as bedroom *v.s.* living room. We reveal that GAN synthesizes the shared objects at some intermediate layers. By controlling their activations only, we can easily overwrite the category of the output image, *e.g.* turning bedroom into living room, while preserve its original layout and high-level attributes such as indoor lighting. Sec.3.3 further shows that our approach can faithfully identify the most relevant attributes associated with a particular scene, facilitating semantic scene manipulation.

**Experimental Setting.** The main experiment is conducted on StyleGAN (Karras et al. (2018b)), but we also extend our analysis to PGGAN (Karras et al. (2018a)) and BigGAN (Brock et al. (2019)). Most models are trained to synthesize scene images within a particular scene category, but we also train a *mixed* StyleGAN model on a collection of images including bedroom, living room, and dining room to better understand how GAN encodes the categorical information and their associated objects. We use *off-the-shelf* image classifiers to assign synthesized scenes with semantic scores, including a layout estimator (Zhang et al. (2019)), a scene category recognizer (Zhou et al. (2017)), and an attribute classifier (Zhou et al. (2017)). We further extract color scheme of a scene image through its hue histogram in HSV space. More details of the GAN models, the image classifiers, and the semantic boundary search process can be found in **Appendix**.

### 3.1 EMERGING SEMANTIC HIERARCHY

Humans typically interpret a scene in a hierarchy of semantics, from its layout, underlying objects, to the detailed attributes and the color scheme. This section will show that GAN composes a scene over

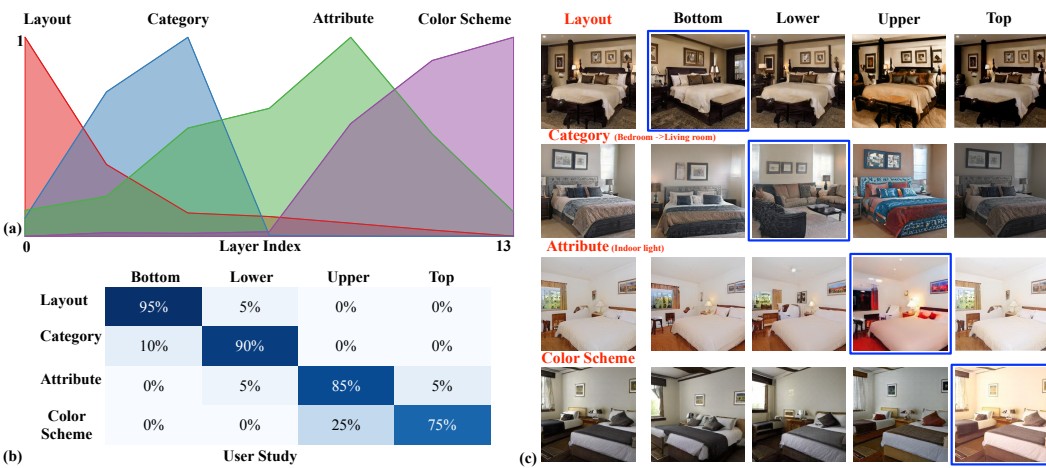

Figure 3: (a) Four levels of visual abstractions emerge at different layers of StyleGAN. Vertical axis shows the normalized perturbation score $\Delta s_i$. (b) User study on how different layers correspond to variation factors from different abstraction levels. (c) Layer-wise manipulation result. The first column is the original synthesized images, and the other columns are the manipulated images at layers from four different stages respectively. Blue boxes highlight the results from varying the latent code at the most proper layers for the target concept.

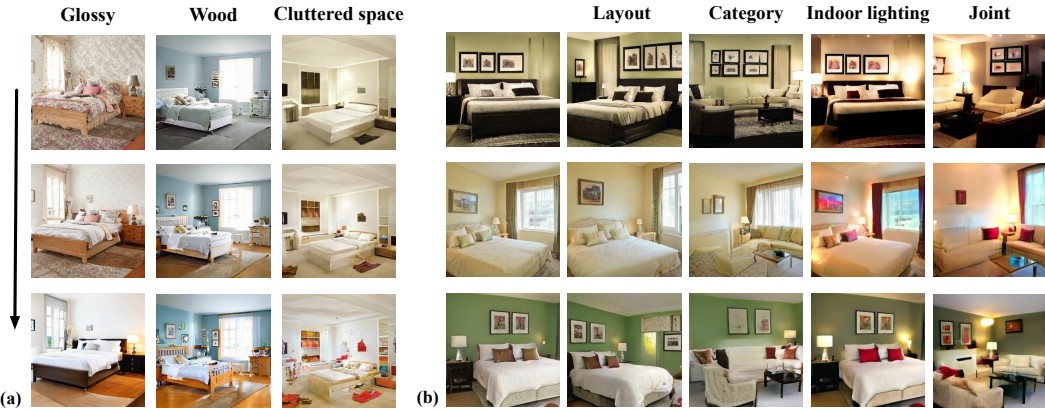

Figure 4: (a) Independent attribute manipulation results on high layers. The middle row are the source images. We are able to both decrease (top row) and increase (bottom row) the variation factors in the images. (b) Joint manipulation results, where the *layout* is manipulated at early layers, the *categorical objects* are manipulated at middle layers, while indoor lighting *attribute* is manipulated at later layers. The first column indicates the source images, the middle three columns are the independently manipulated images.

the layers in a similar way with human perception. To enable analysis on layout and category, we take the *mixed* StyleGAN model trained on indoor scenes as the target model. StyleGAN (Karras et al. (2018b)) learns a more disentangled latent space $\mathcal{W}$ on top of the conventional latent space $\mathcal{Z}$. Besides, StyleGAN feeds the latent code $\mathbf{w} \in \mathcal{W}$ to each convolutional layer with different transformations instead of only feeding it to the first layer. Specifically, for $\ell$-th layer, $\mathbf{w}$ is linearly transformed to layer-wise transformed latent code $\mathbf{y}^{(\ell)}$ with $\mathbf{y}^{(\ell)} = \mathbf{A}^{(\ell)}\mathbf{w} + \mathbf{b}^{(\ell)}$, where $\mathbf{A}^{(\ell)}, \mathbf{b}^{(\ell)}$ are the weight and bias for style transformation respectively. We thus perform layer-wise analysis by studying $\mathbf{y}^{(\ell)}$ instead of $\mathbf{z}$ in Eq.(1).

To quantify the importance of each layer with respect to each variation factor, we use the re-scoring technique to identify the causality between the layer-wise generative representation $\mathbf{y}^{(\ell)}$ and the semantic emergence. The normalized score in Fig.3(a) shows that the layers of the generator in GAN are specialized to compose semantics in a hierarchical manner: the bottom layers determine the layout, the lower layers and upper layers control category-level and attribute-level variations respectively, while color scheme is mostly rendered at the top. This is consistent with human perception.

To visually inspect the identified variation factors, we move latent vector along the boundaries at different layers to show how the synthesis varies correspondingly. For example, given a boundary in regard to room layout, we vary the latent code towards the normal direction at bottom, lower, upper,

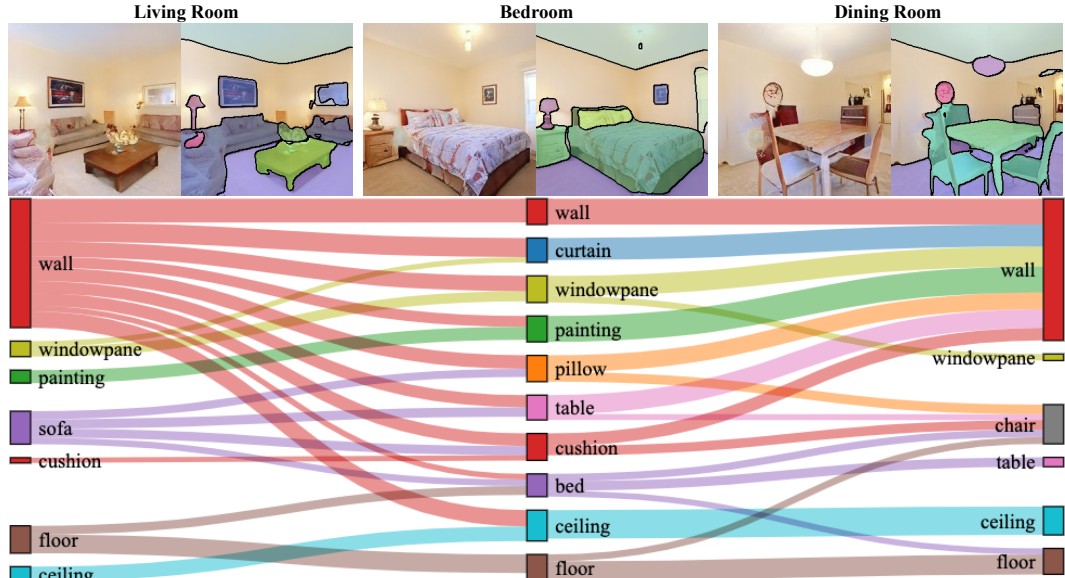

Figure 5: Objects are transformed by GAN to represent different scene categories. On the top shows that the object segmentation mask varies when manipulating a living room to bedroom, and further to dining room. On the bottom visualizes the object mapping that appears during category transition, where pixels are counted only from object level instead of instance level. GAN is able to learn shared objects as well as the transformation of objects with similar appearance when trained to synthesize scene images from more than one category.

and top layers respectively. Fig.3(c) shows the qualitative results for several concepts. We see that the emerged variation factors follow a highly-structured semantic hierarchy, *e.g.*, layout can be best controlled at early stage while color scheme can only be changed at final stage. Besides, varying latent code at the inappropriate layers may also change the image content, but the changing might be inconsistent with the desired output. For example, in the second row, modulating the code at bottom layers for category only leads to a random change in the scene viewpoint.

To better evaluate the manipulability across layers, we conduct a user study. We first generate 500 samples and manipulate them with respect to several concepts on different layers. For each concept, 20 users are asked to choose the most appropriate layers for manipulation. Fig.3(b) shows the user study results, where most people think bottom layers best align with layout, lower layers control scene category, *etc*. This is consistent with our observations in Fig.3(a) and (c). It suggests that hierarchical variation factors emerge inside the generative representation for synthesizing scenes. and that our re-scoring method indeed helps identify the variation factors from a broad set of semantics.

Identifying the semantic hierarchy and the variation factors across layers facilitates semantic scene manipulation. We can simply push the latent code toward the boundary of the desired attribute at the appropriate layer. Fig.4(a) shows that we can change the decoration style (crude to glossy), the material of furniture (cloth to wood), or even the cleanliness (tidy to cluttered) respectively. Furthermore, we can jointly manipulate the hierarchical variation factors. In Fig.4(b) we simultaneously change the room layout (rotating viewpoint) at early layers, scene category (converting bedroom to living room) at middle layers, and scene attribute (increasing indoor lighting) at later layers.

## 3.2 WHAT MAKES A SCENE?

As mentioned above, GAN models for synthesizing scenes are capable of encoding hierarchical semantics inside the generative representation, *i.e.*, from layout, category, to scene attribute and color scheme. One of the most noticeable properties is that the middle layers of GAN actually synthesize different objects for different scene categories. It raises the question on what makes a scene as living room rather than bedroom. Thus we further dive into the encoding of categorical information in GANs, to quantify how GAN interprets a scene category as well as how the scene category is transformed from object perspective.

We employ the StyleGAN model trained on the mixture of bedroom, living room, and dining room, and then search the semantic boundary between each two categories. To extract the objects from

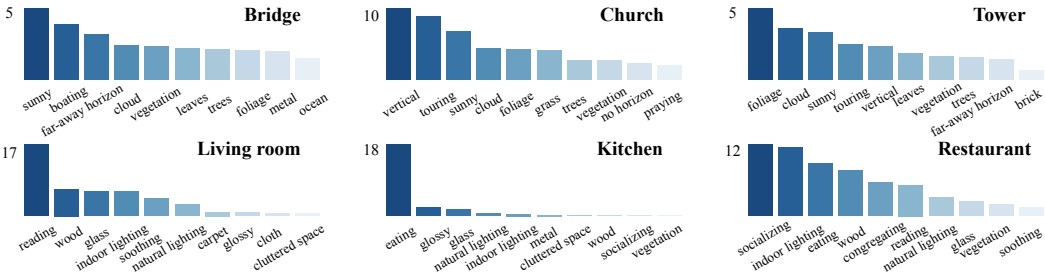

Figure 6: Comparison of the top scene attributes identified in the generative representations learned by StyleGAN models for synthesizing different scenes. Vertical axis shows the perturbation score $\Delta s_i$.

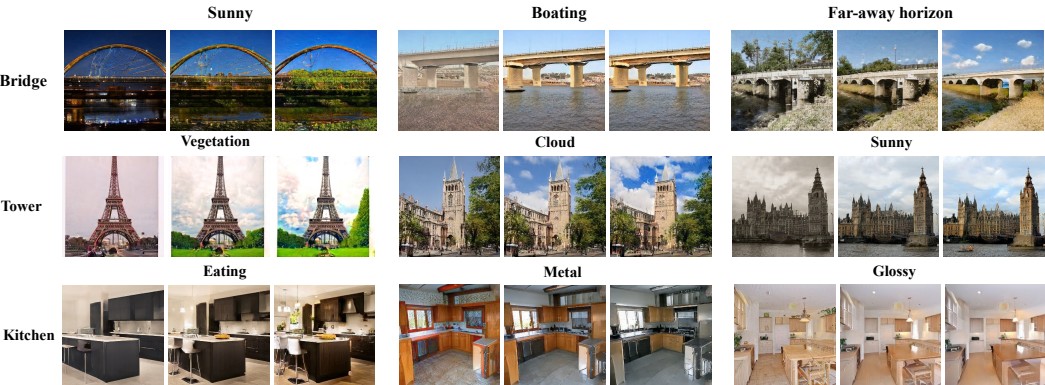

Figure 7: Manipulation results on StyleGAN models trained for synthesizing different scenes. For each triple, on top shows the target attribute, the first image is the source image, the other two images are generated by increasing the manipulation magnitude. Please see the demo video for continuous manipulation via this link.

the synthesized images, we apply a semantic segmentation model (Xiao et al. (2018b)), which can segment 150 objects (tv, sofa, *etc*) and stuff (ceiling, floor, *etc*). Specifically, we first randomly synthesize 500 living room images, and then vary the corresponding latent codes towards the "living room-bedroom" boundary and "bedroom-dining room" boundary in turn. We segment the images before and after manipulation to get the segmentation masks, as shown in Fig.5. After tracking label mapping for each pixel via the image coordinate during the manipulation process, we are able to compute the statistics on how objects are transformed along with category changing.

Fig.5 shows the objects mapping in the category transformation process. We can see that (1) When image is manipulated among different categories, most of the stuff classes (*e.g.*, ceiling and floor) remain the same, but some objects are mapped into other classes. For example, sofa in living room is mapped to pillow and bed in bedroom, and bed in bedroom is further mapped to table and chair in dining room. This phenomenon happens because sofa, bed, dining table and chair are distinguishable objects for living room, bedroom, and dining room respectively. (2) Some objects are sharable between different scene categories, and the GAN model is able to spot such property and learn to generate these shared objects across different classes. For example, the lamp in living room (on the left boundary of the image) still remains after the image is converted to bedroom. (3) With the ability to learn object mapping as well as share objects across different classes, we are able to turn an unconditional GAN into a GAN that can control category. Typically, to make GAN produce images from different categories, class labels have to be fed into the generator to learn a categorical embedding, like BigGAN (Brock et al. (2019)). Our result suggests an alternative approach.

### 3.3 DIVERSE ATTRIBUTE MANIPULATION

The emergence of variation factors for scene synthesis depends on the training data. Here we apply our method to a collection of StyleGAN models, to capture a wide range of manipulatable attributes out of the 102 scene attributes we use. Each styleGAN in the collection is trained to synthesize scene images from a certain category, including both outdoor (bridge, church, tower) and indoor scenes (living room, kitchen).

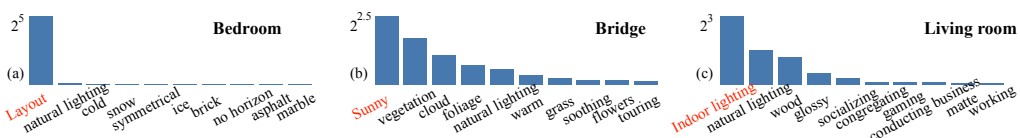

Figure 8: Effects on scene attributes (already sorted) when varying a particular variation factor (in red color). Vertical axis shows the perturbation score $\Delta s_i$ in log scale.

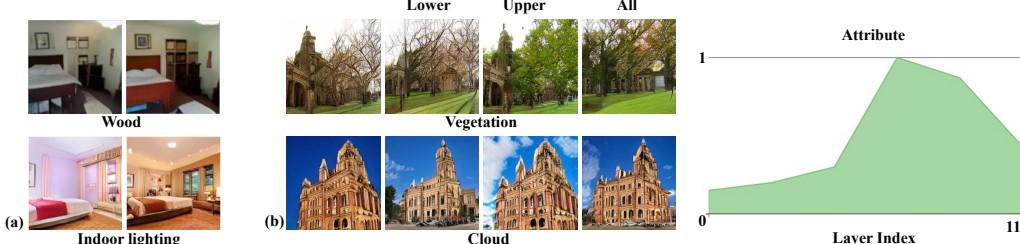

Figure 9: (a) Some variation factors identified from PGGAN (bedroom). (b) Layer-wise analysis on BigGAN from attribute level.

Fig.6 shows the top-10 relevant semantics to each model. We can see that "sunny" has high scores on all outdoor categories, while "lighting" has high scores on all indoor categories. Furthermore, "boating" is identified for bridge model, "touring" for church and tower, "reading" for living room, "eating" for kitchen, and "socializing" for restaurant. These results are highly consistent with human perception, suggesting the effectiveness of the proposed quantification method. Fig.7 further shows manipulation results with respect to the scene attributes identified by our approach. We realistically manipulate the synthesized image with desired semantics. More results can be found in **Appendix**.

## 4 DISCUSSION AND CONCLUSION

**Disentanglement of Semantics.** Some variation factors we detect in the generative representation are more disentangled with each other than other semantics. Compared to the perceptual path length and linear separability described in Karras et al. (2018b) and the cosine similarity proposed in Shen et al. (2019), our work offers a new metric for disentanglement analysis. In particular, we move the latent code along one semantic direction and then check how the semantic scores of other factors change accordingly. As shown in Fig.8(a), when we modify the spatial layout, all scene attributes are barely affected, suggesting that GAN learns to disentangle layout-level semantic from attribute-level. However, there are also some scene attributes (from same abstraction level) entangling with each other. Taking Fig.8(c) as an example, when modulating "indoor lighting", "natural lighting" also varies. This is also aligned with human perception, further demonstrating the effectiveness of our proposed quantification metric.

**Application to Other GANs.** We further apply our method for two other GAN structures, *i.e.*, PGGAN (Karras et al. (2018a)) and BigGAN (Brock et al. (2019)). These two models are trained on LSUN dataset (Yu et al. (2015)) and Places dataset (Zhou et al. (2017)) respectively. Compared to StyleGAN, PGGAN feeds the latent vector only to the very first convolutional layer and hence does not support layer-wise analysis. But the proposed re-scoring method can still be applied to help identify manipulatable semantics, as shown in Fig.9(a). BigGAN is the state-of-the-art conditional GAN model that concatenates the latent vector with a class-guided embedding code before feeding it to the generator, and it also allows layer-wise analysis like StyleGAN. Fig.9(b) gives analysis results on BigGAN from attribute level, where we can tell that scene attribute can be best modified at upper layers compared to lower layers or all layers. Meanwhile, the quantitative curve shows consistent result with the discovery on StyleGAN as in Fig.3(a). These results demonstrate the generalization ability of our approach as well as the emergence of manipulatable factors in other GANs.

In this paper, we show the emergence of highly-structured variation factors inside the deep generative representations learned by GANs with layer-wise stochasticity. In particular, the GAN model spontaneously learns to set up layout at early layers, generate categorical objects at middle layers, and render scene attribute and color scheme at later layers when trained to synthesize scenes. A re-scoring method is proposed to quantitatively identify the manipulatable semantic concepts within a well-trained model, enabling photo-realistic scene manipulation.

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

APPENDIX

Sec.A introduces the implementation details, including the GAN models used in this work, the *off-the-shelf* classifiers used for semantic score prediction, and the process of semantic identification. Sec.B contains the ablation study for showing why the proposed re-scoring technique is essential for identifying variation factors in GAN. Sec.C discusses the limitation of our method as well as some future directions. Sec.D contains more semantic scene manipulation results for a wide range of scene categories and concepts. Sec.E shows the details model structures of StyleGAN and BigGAN, both of which employ layer-wise latent codes. Sec.F provides the ablation study on layer-wise manipulation from different abstraction levels.

## A    IMPLEMENTATION DETAILS

### A.1    GAN MODELS

We conduct experiments on three state-of-the-art generative models, including PGGAN (Karras et al. (2018a)), StyleGAN (Karras et al. (2018b)), and BigGAN (Brock et al. (2019)). Among them, PGGAN and StyleGAN are trained on LSUN dataset (Yu et al. (2015)) while BigGAN is trained on Places dataset (Zhou et al. (2017)). LSUN dataset consists of 7 indoor scene categories and 3 outdoor scene categories, and Places dataset contains 10 million images across 434 categories. For PGGAN model, we use the officially released models[2], each of which is trained to synthesize scene within a particular category of LSUN dataset. For StyleGAN, only one model related to scene synthesis (*i.e.*, bedroom) is released[3]. For a more thorough analysis, we use the official implementation[4] to train some additional models on other scene categories, including both indoor scenes (living room, kitchen, restaurant) and out door scenes (bridge, church, tower). We also train a *mixed* model on the combination of images from bedroom, living room, and dining room with same implementation. This model is specifically used for categorical analysis. For each StyleGAN model, Tab.1 shows the category, the number of training samples, as well as the corresponding Fréchet inception distances (FID) (Heusel et al. (2017)) which can reflect the synthesis quality to some extent. For BigGAN, we use the author's officially unofficial PyTorch BigGAN implementation[5] to train a conditional generative model by taking category label as constraint. The resolution of the scene images synthesized by all of the above models is $256 \times 256$.

Table 1: Description of the StyleGAN models trained on different categories.

| Scene Category | Indoor / Outdoor | Training Samples | FID (lower is better) |
|---|---|---|---|
| bedroom (official) | Indoor | 3M | 2.65 |
| living room | Indoor | 1.3M | 5.16 |
| kitchen | Indoor | 1M | 5.06 |
| restaurant | Indoor | 626K | 4.03 |
| bridge | Outdoor | 819K | 6.42 |
| church | Outdoor | 126K | 4.82 |
| tower | Outdoor | 708K | 5.99 |
| Mixed | Indoor | 500K each | 3.74 |

### A.2    SEMANTIC CLASSIFIERS

To extract semantic from synthesized images, we employ some *off-the-shelf* image classifiers to assign these images with semantic scores from multiple abstraction levels, including *layout*, *category*, *scene attribute*, and *color scheme*. Specifically, we use (1) a *layout estimator* (Zhang et al. (2019)), which is able to predict the spatial structure of a indoor place, (2) a *scene category classifier* (Zhou et al. (2017)), which is able to classify a scene image to 365 categories, and (3) an *attribute predictor*

---

[2]These PGGAN models can be found at `https://drive.google.com/open?id=15hvzxt_XxuokSmj0uO4xxMTMWVc0cIMU`.

[3]The StyleGAN model can be found at `https://drive.google.com/drive/folders/1MASQyN5m0voPcx7-9K0r5gObhvvPups7`.

[4]The implementation of StyleGAN can be found at `https://github.com/NVlabs/stylegan`.

[5]The implementation of BigGAN can be found at `https://github.com/ajbrock/BigGAN-PyTorch`.

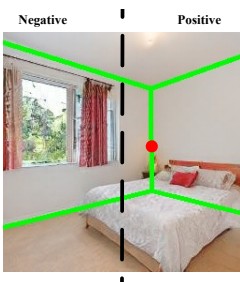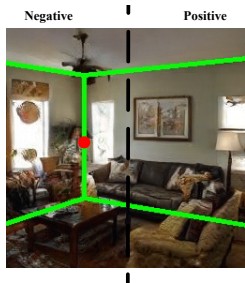

Figure 10: The definition of layout for indoor scenes. Green lines represent for the outline prediction from the layout estimator. The dashed line indicates the horizontal center, and the red point is the center point of the intersection line of two walls. The relative position between the vertical line and the center point is used to split the dataset. For example, image on the left is treated as positive sample, while the one on the right is treated as negative sample.

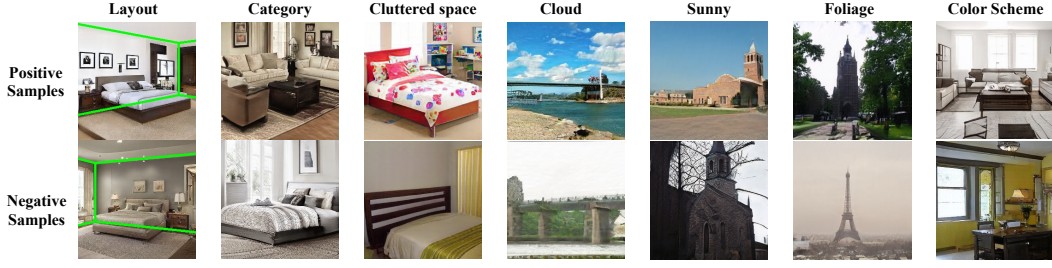

Figure 11: Samples for training decision boundary with respect to layout, scene category, and various scene attributes.

(Zhou et al. (2017)), which is capable of predicting 102 pre-defined scene attributes (*e.g.*, sunny and dirty). We also extract color scheme of a scene image through its hue histogram in HSV space. Among them, the category classifier and attribute predictor can directly output the probability of how likely an image belongs to a certain category or how likely an image has a particular attribute. As for the layout estimator, it only detects the outline structure of a indoor place, shown as the green line in Fig.10.

### A.3 SEMANTIC PROBING AND VERIFICATION

Given a well-trained GAN model for analysis, we first generate a collection of synthesized scene images by randomly sampling $N$ latent codes. To ensure capturing all the potential variation factors, we set $N = 500,000$. We then use the aforementioned image classifiers to assign semantic scores for each visual concept. It is worth noting that we use the relative position between image horizontal center and the intersection line of two walls to quantify layout, as shown in Fig.10. After that, for each candidate, we select $2,000$ images with the highest response as positive samples, and another $2,000$ with the lowest response as negative ones. Fig.11 shows some examples, where living room and bedroom are treated as positive and negative for scene category respectively. We then train a linear SVM by treating it as a bi-classification problem (*i.e.*, data is the sampled latent code while label is binary indicating whether the target semantic appears in the corresponding synthesis or not) to get a linear decision boundary. Finally, we re-generate $K = 1,000$ samples for semantic verification as described in Sec.2.2.

### B ABLATION STUDY ON RE-SCORING TECHNIQUE

Before performing the proposed re-scoring technique, we have two more steps, which are (1) assigning semantic scores for synthesized samples, and (2) training SVM classifiers to search semantic boundary. We would like to verify the essentiality of the re-scoring technique in identifying manipulatable semantics. We conduct ablation study on the StyleGAN model trained for synthesizing bedrooms. As shown in Fig.12, the left figure sorts the scene attributes by how many samples are labeled as positive

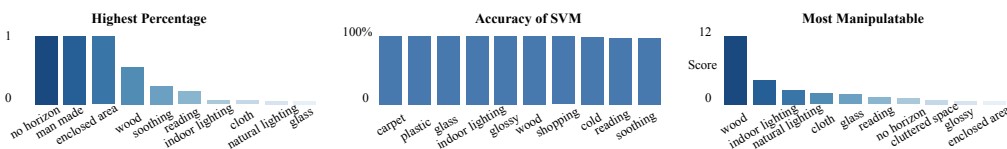

Figure 12: Ablation study on the proposed re-scoring technique with StyleGAN model for bedroom synthesis.

ones, the middle figure sorts by the accuracy of the trained SVM classifiers, while the right figure sorts by our proposed quantification metric.

In left figure, "no horizon", "man-made", and "enclosed area" are attributes with highest percentage. However, all these three attributes are default properties of bedroom and thus not manipulatable. On the contrary, with the re-scoring technique for verification, our method successfully filters out these invariable candidates and reveals more meaningful semantics, like "wood" and "indoor lighting". In addition, our method also manages to identify some less frequent but actually manipulatable scene attributes, such as "cluttered space".

In middle figure, almost all attributes get similar scores, making them indistinguishable. Actually, even the worst SVM classifier (*i.e.*, "railroad") achieves 72.3% accuracy. That is because even some variation factors are not encoded in the latent representation (or say, not manipulatable), the corresponding attribute classifier still assign synthesized images with different scores. Training SVM on these inaccurate data can also result in a separation boundary, even it is not expected as the target concept. Therefore, only relying on the SVM classifier is not enough to detect relevant variation factors. By contrast, our method pays more attention to the score modulation after varying the latent code, which is not biased by the initial response of attribute classifier or the performance of SVM. As a result, we are able to thoroughly yet precisely detect the variation factors in the latent space from a broad candidate set.

## C  LIMITATION AND FUTURE WORK

Despite the success of our proposed re-scoring technique in quantitatively identifying the hierarchical manipulatable latent variation factors in the deep generative representations, there are several limitations for future improvement.

First, the layout classifier can only detect the layout structure of indoor scenes. But for a more general analysis on both indoor and outdoor scene categories, there lacks of an unified definition of the spatial layout. For example, our framework cannot change the layout of outdoor church images. In future work, we will leverage the computational photography tools that recover the 3D camera pose of the image, thus we can extract more universal viewpoint representation for the synthesized images. Second, our proposed re-scoring technique relies on the performances of the off-the-shelf classifiers. For some of the attributes, the classifiers are not so accurate, which leads to poor manipulation boundary. This problem could be addressed with more powerful discriminative models. Third, for simplicity we only use the linear SVM for semantic boundary search. This limits our framework from interpreting the latent semantic subspace with more complex and nonlinear structure.

## D  MANIPULATION OF SYNTHESIZED SCENES

Our proposed method can not only identify hierarchical variation factors from learned generative representation, but futher facilitate semantic scene manipulation. Fig.13 shows the manipulation results from *layout* level and *category* level. Fig.14 and Fig.15 show the manipulation results from *attribute* level on indoor scenes and outdoor scenes respectively. Fig.16 shows the joint manipulation by modulating the latent code along the direction of desired semantics *at the most appropriate layer*. All experiments are conducted on StyleGAN model.

**Layout**

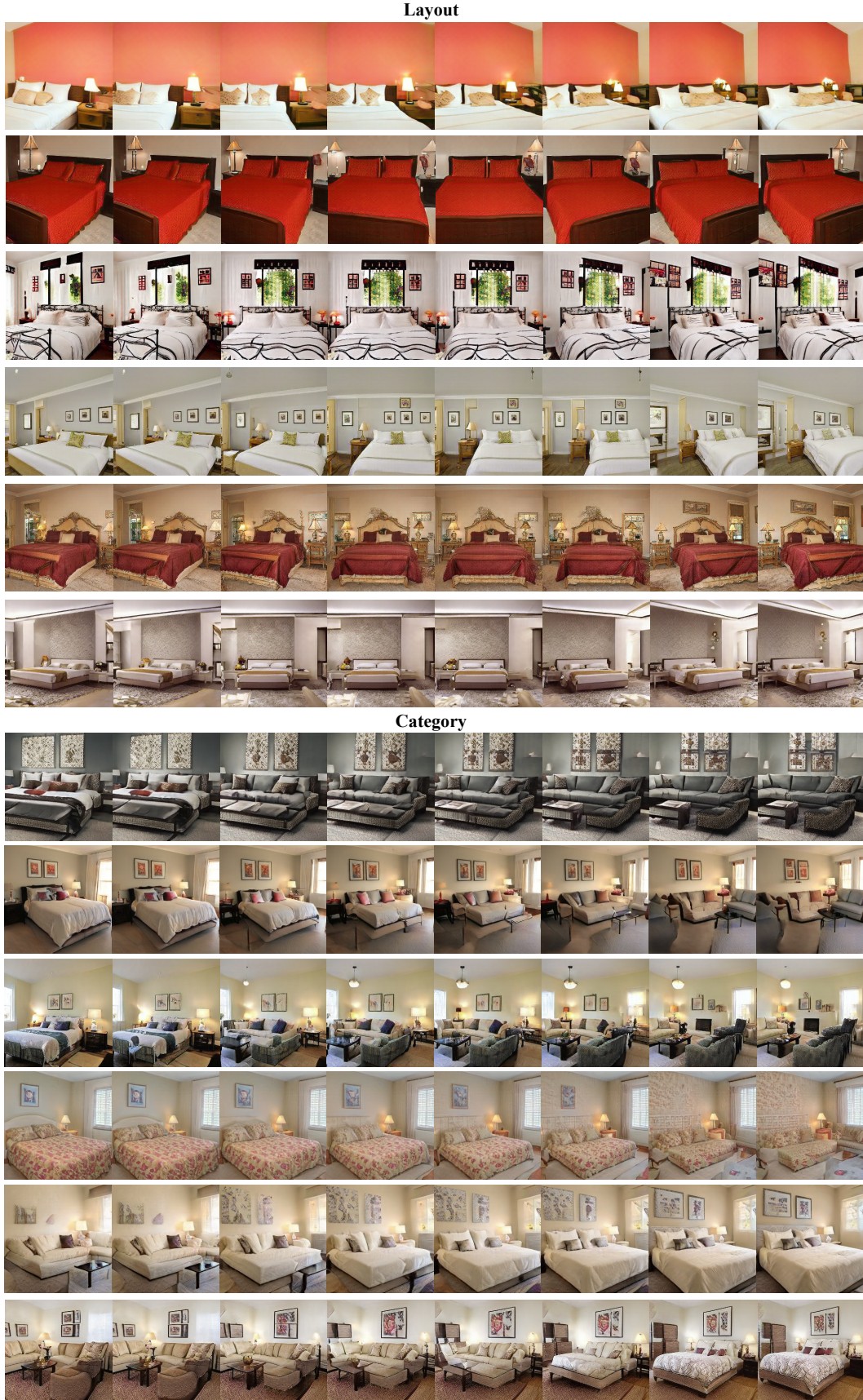

**Category**

Figure 13: Layout and category manipulation results.

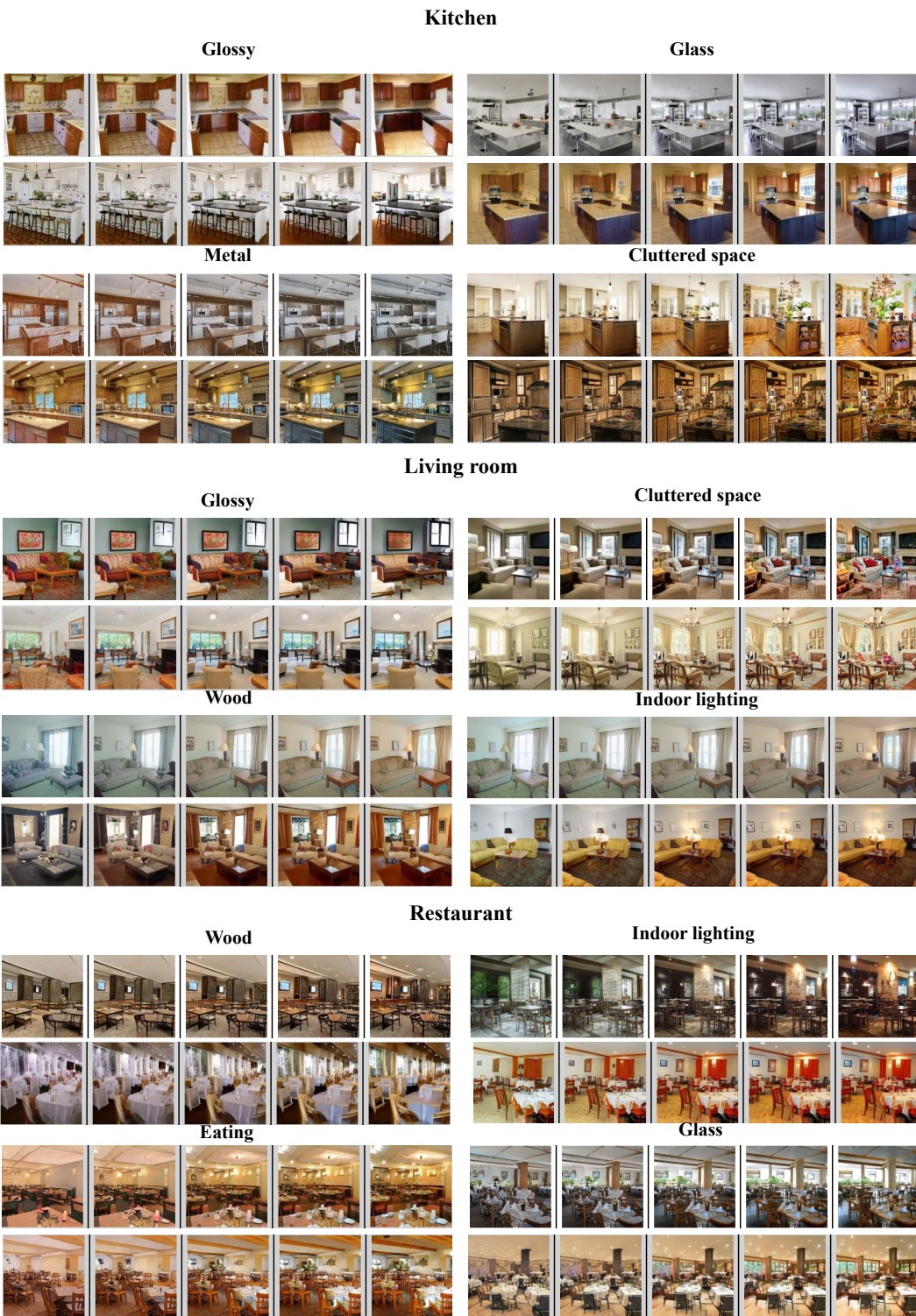

Figure 14: Manipulating the attributes of indoor scenes at different scores (low to high).

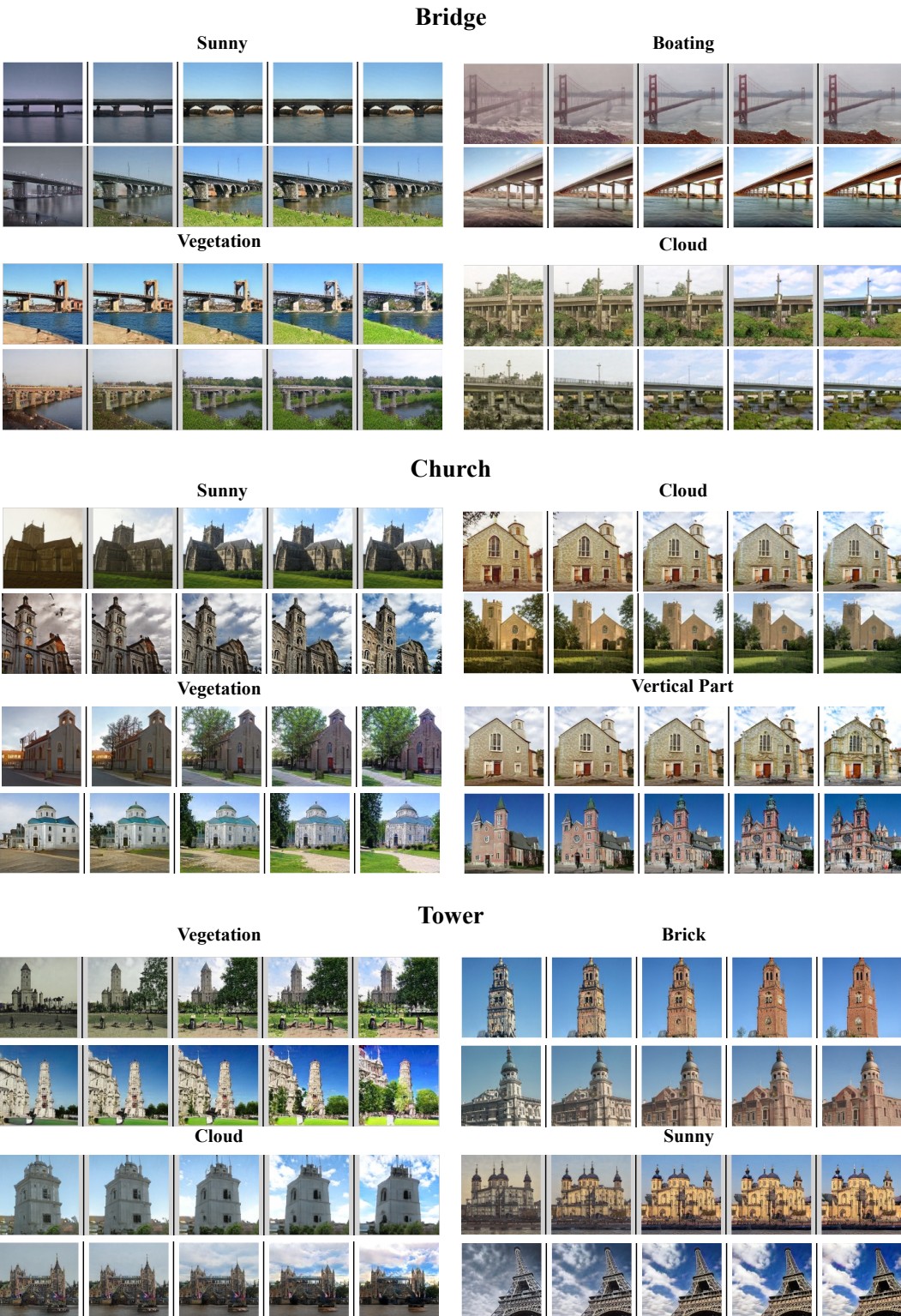

Figure 15: Manipulating the attributes of outdoor scenes at different scores (low to high).

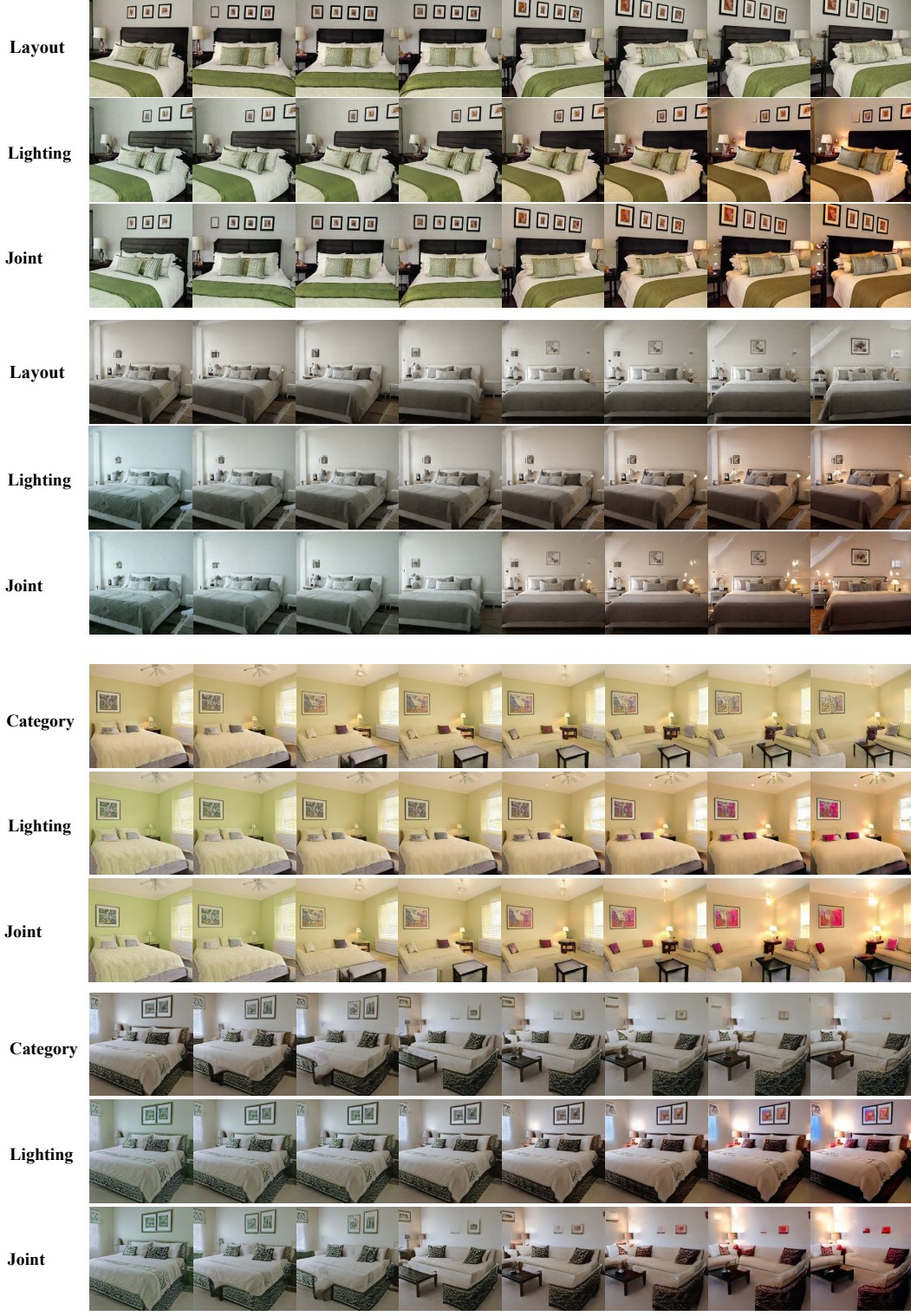

Figure 16: Independent and joint manipulation results.

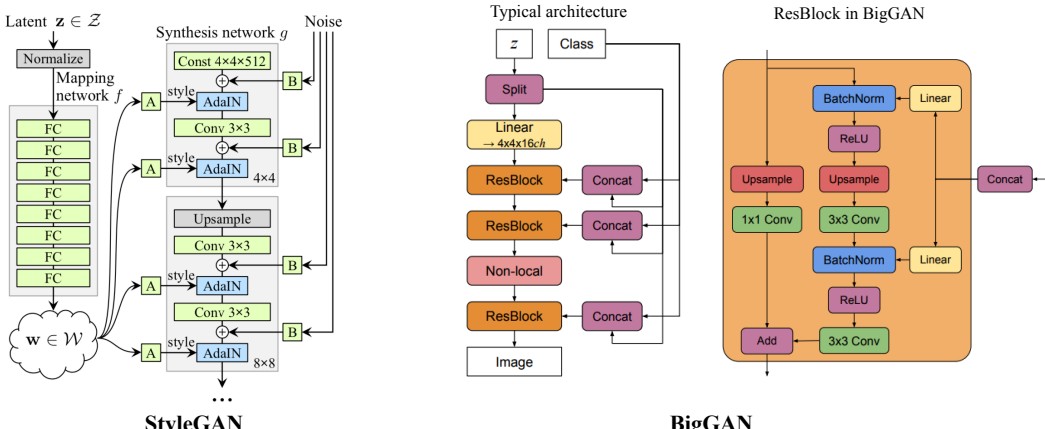

Figure 17: Generator architectures of StyleGAN (Karras et al. (2018b)) and BigGAN (Brock et al. (2019)), both of which introduces layer-wise stochasticity (*i.e.*, latent codes are fed into all convolutional layers instead of only the first layer). Note that the diagrams are borrowed from the original papers.

# E GENERATOR STRUCTURES

This work conducts experiments on state-of-the-art deep generative models for high-resolution scene synthesis, including StyleGAN (Karras et al. (2018b)), BigGAN (Brock et al. (2019)), and PGGAN (Karras et al. (2018a)). Among them, PGGAN employs the conventional generator structure where the latent code is only fed into the very first layer. Differently, StyleGAN and BigGAN introduces layer-wise stochasticity by feeding latent codes to all convolutional layers as shown in Fig.17. It is worth mentioning that more and more latest GAN models inherit the design of using layer-wise latent codes to achieve better generation quality, such as SinGAN (Shaham et al. (2019)) and HoloGAN (Nguyen-Phuoc et al. (2019)). And our layer-wise analysis sheds light on why it is effective.

In StyleGAN model that is trained to produce $256 \times 256$ scene images, there are totally 14 convolutional layers. According to our experimental results, *layout*, *category*, *attribute*, *color scheme* correspond to *bottom*, *lower*, *upper*, and *top* layers respectively, which are actually $[0, 2)$, $[2, 6)$, $[6, 12)$ and $[12, 14)$ layers. As for BigGAN model with $256 \times 256$ resolution, there are total 12 convolutional layers. As the category information is already encoded in the "class" code as shown in Fig.17, we only separate the layers to two groups, which are *lower* (bottom 6 layers) and *upper* (top 6 layers). Since our layout model can only be applied to indoor scenes yet the BigGAN model is trained on Places dataset which contains both indoor and outdoor categories, we only analyze BigGAN from attribute level as shown in Fig.9(b). Both visualization results and quantitative curve suggest that attribute-level semantics are better controlled by *upper* layers of BigGAN. For example, when manipulating "vegetation" attribute at lower layers or all layers, the spatial information varies unexpectedly, while manipulating at upper layers gives desired output.

**Upper**     **All**                     **Upper**     **All**

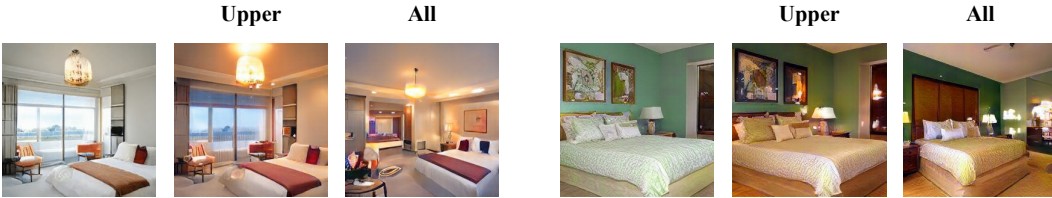

Figure 18: Comparison results between manipulating latent codes at only upper (attribute-relevant) layers and manipulating latent codes at all layers with respect to *indoor lighting* on StyleGAN.

**Layout**       **Category**       **Indoor lighting**       **Color Scheme**

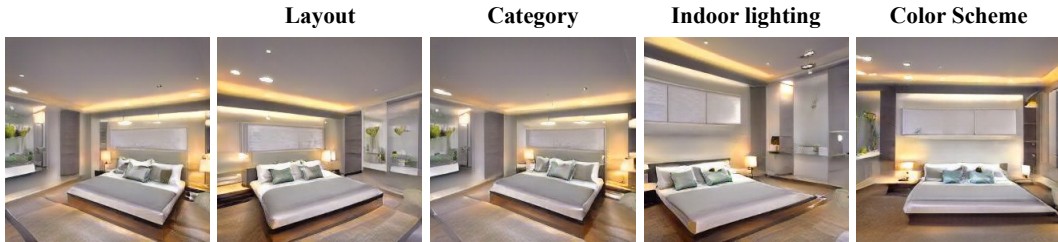

Figure 19: Manipulation at the *bottom* layers in 4 different directions, including *layout*, *category*, *indoor lighting*, and *color scheme* on StyleGAN.

## F  ABLATION STUDY ON LAYER-WISE MANIPULATION

To further validate the emergence of semantic hierarchy, we make ablation study on layer-wise manipulation with StyleGAN model.

First, we select "indoor lighting" as the target semantic, and vary the latent code only on upper (attribute-relevant) layers *v.s.* on all layers. We can easily tell from Fig.18 that when manipulation "indoor lighting" at all layers, the objects inside the room are also changed. By contrast, manipulating latent codes only at attribute-relevant layers can satisfyingly increase the indoor lighting without affecting other factors.

Second, we select bottom layers as the target layers, and select boundaries from all four abstraction levels for manipulation. As shown in Fig.19, no matter what level of semantics we choose, as long as the latent code is modified at bottom (layout-relevant) layers, only layout instead of all other semantics varies.

These two experiments further verify our discovery about the emergence of semantic hierarchy.

