# OpenReview forum: "Semantic Hierarchy Emerges in the Deep Generative Representations for Scene Synthesis"
_ICLR.cc/2020/Conference — Reject_

### Official Review · AnonReviewer1 · 2019-10-25
**Official Blind Review #1**

**Rating:** 6

**Review:**

Updates after author response:
I'd like to thank the authors for their detailed responses. Some of my primary concerns were regarding the presentation, and I feel they have been mostly addressed with the changes to the introduction and abstract (I'd still recommend using 'layerwise latent code' instead of 'layerwise representation' everywhere in the text). The additional qualitative results showing the benefits of manipulating 'z' vs y_l were also helpful. Finally, I agree that given the popularity of StyleGAN like models, the investigation methodology proposed, and the insights presented might be useful to a broad audience. Overall, I am inclined to update my rating to lean towards acceptance.

---------------------------
This paper investigates the aspects encoded by the latent variables input to different layers in StyleGAN (Karras et. al.), and demonstrates that these correspond to encoding different aspects of the scene across layers e.g. initial ones correspond to layout, final ones to lighting.

The ’StyleGAN’ work first-generates a per-layer latent code y_l (from a global latent variable w), and uses these in a generative model. This paper investigates which layer’s latent codes best explain certain variations in scenes. To formalize the notion of how a latent vector is causally related to a scene property, the approach here is to use an off-the-shelf classifier for the property, and a) find a linear decision boundary in the latent space, and b) quantifying whether changing the latent code indeed affects the predicted score.

Positives:
1. The analysis presented in the work is thorough and results interesting. The paper analyzes the relation of various scene properties w.r.t the latent variables across layers, and does convincingly show that aspects like layout, category, attribute etc, are related to different layers.

2. The visual results depicting manipulation of specific properties of scenes by changing specific variables in the latent space, and the ones in Sec 3.2 studying transitions across scene types, are also impressive and interesting.

3. The proposed way of measuring the ‘manipulability’ of an aspect of a scene w.r.t a latent variable is simple and elegant, thought I have some concerns regarding its general applicability (see below).

Despite these positives, I am not sure about accepting the paper because I feel the investigation methods and the results are both very specific to a particular sort of GAN, and the writing (introduction, abstract, related work etc.) pitch the paper as being more general than it is, and claim the insights to be more applicable. More specifically:

1) The text claims the approach ‘probes the layer-wise representations’. However, what is actually investigated is the layer-wise latent code (NOT ‘representation’ which is typically defined to mean the responses of filters/outputs of each layer). In fact, I do not think this work is directly applicable to probing ‘representations’ as the term is normally used because it may be too high-dimensional to infer meaningful linear decision boundaries, or directly manipulate it.

2) All the initial text in the paper’s abstract, introduction etc. leads the reader to believe that the findings here are generally applicable e.g. the sentence “the generative representations learned by GAN are specialized to synthesize different hierarchical semantics” should actually be something like “the per-layer latent variables for StyleGAN affect different levels of scene semantics“. Independent of any other concerns, I would be hesitant to accept the paper with the current writing given the very general nature of assertions made despite experiments in far more specific settings.

3) In Sec 4, this paper only shows some sample results other models e.g. BIGGAN, but no ’semantic hierarchy in deep generative representation’ is shown (not surprising given only a global latent code).  As the discussion also alludes to, I do not think this approach would yield any insights if a GAN does not have a multi-layered latent code.

4) Finally, while the results obtained for StyleGAN do convincingly show the causal relations claimed, these results are essentially backing up the insights that led to the design of StyleGAN i.e. having a single-level latent variable capture all source of variation is sub-optimal.

5) This is not a really weakness, but perhaps an ablation that may help. The results showing scene property manipulation e.g. in Fig 4 are obtained by varying a certain y_l, and it’d help to also show the results if the initial latent code w was modified directly (therefore affecting all layers!). It would be interesting to know if this adversely affects constancy of some aspects e.g. maybe objects also change in addition to layout.

Overall, while the results are interesting, they are only in context of a specific GAN, and using an approach that is applicable to generative models having a multi-layer code. I feel the paper should also be written better to be more precise regarding the claims. While the rating here only allows me to give a ‘3’ as a weak reject, I am perhaps a bit more towards borderline (though leaning towards reject) than that indicates.

**Experience Assessment:**

I have read many papers in this area.

**Review Assessment: Checking Correctness Of Derivations And Theory:**

I assessed the sensibility of the derivations and theory.

**Review Assessment: Checking Correctness Of Experiments:**

I carefully checked the experiments.

**Review Assessment: Thoroughness In Paper Reading:**

I read the paper thoroughly.

---

> ### Author Response · Authors · 2019-11-09
> **Response to Review #1**
>
> Thanks for the valuable comments.
>
> Q1: "what is actually investigated is the layer-wise latent code (NOT ‘representation’ which is typically defined to mean the responses of filters/outputs of each layer)"
> A1: Unlike the classification networks, where the output of each layer can be considered as the abstract feature (or say, representation) of the original input image, deep generative model learns to map the pre-defined latent distribution to observed image distribution. In the image generation process, the deep representation at each layer (especially for StyleGAN and BigGAN) is actually directly derived from the projected latent code at each layer. Therefore, we consider the latent code as the "generative representation", which may be slightly different from the conventional definition in the classification networks. Furthermore, since the GAN model is fixed, the input latent code and the output response of filters are implicitly causal. From this perspective, studying the latent code is equivalent to studying the layer output to some extent. We clarify this definition in the updated submission.
>
> Q2: "All the initial text in the paper’s abstract, introduction etc. leads the reader to believe that the findings here are generally applicable"
> A2: Thanks for the suggestion. We tone down the claim in abstract and introduction in the updated submission. We conduct the layer-wise analysis on StyleGAN and BigGAN, but the proposed probing and manipulation technique can be generalized to other GANs with single latent code as well, such as our experiment on PGGAN.
>
> Q3: "this paper only shows some sample results other models e.g. BIGGAN"
> A3: First, to our knowledge, StyleGAN is currently the best deep generative model for high-resolution scene synthesis. That is why we mainly conduct experiments on StyleGAN structure. According to the experimental results, we believe the reason that StyleGAN achieves such good generation quality is due to the design of layer-wise latent code. Based on this design, generator can learn different levels of semantics on different layers instead of only the first layer seeing the latent code. Besides, nowadays, more and more latest GAN models inherit this design of using layer-wise latent codes, such as recent ICCV’19 work SinGAN [1] and HoloGAN [2]. As "multi-layered stochasticity" design becomes widely adopted in GANs, our layer-wise analysis sheds light on why it is effective. Also, there may be a misunderstanding about the experiments on BigGAN. According to [3], BigGAN also employs layer-wise latent code, and the BigGAN experiments shown in Fig.9 of the original submission are the results by only modifying the attribute-level latent codes (i.e., upper layers). To make this clear, we update Fig.9 by adding a comparison experiment between modifying all latent codes and modifying only the relevant latent codes, as well as adding a layer-wise analysis on BigGAN. Hope this can address your concern. In addition, the proposed re-scoring technique can also be applied to conventional GAN structures, like PGGAN, which is another state-of-the-art GAN model for scene synthesis yet with single latent code. Sec.4 of the submission shows the results. Indeed, we cannot make layer-wise analysis on PGGAN to show which layer learns which semantic, but we do convincingly find manipulatable semantics in the input latent space. Feel free to name other GAN architectures for high-resolution image synthesis and we are happy to do analysis on them as well.
>
> [1] SinGAN: Learning a Generative Model from a Single Natural Image. Shaham et al., ICCV'19.
> [2] HoloGAN: Unsupervised Learning of 3D Representations From Natural Images. Nguyen-Phuoc et al., ICCV'19.
> [3] Large Scale GAN Training for High Fidelity Natural Image Synthesis. Brock el al., ICLR'19.
>
> Q4: "these results are essentially backing up the insights that led to the design of StyleGAN"
> A4: The design of StyleGAN indeed allows the model to learn a more disentangled representation compared to using a single-level latent variable. However, this fact does not weaken the contributions of this work. First, we propose re-scoring method to identity manipulatable semantics of a given GAN model. Second, we classify scene representation into layout, category (object), attribute, and color scheme four levels to align with human perception, and further study how StyleGAN learns these semantics layer by layer. Such interpretation and understanding of the deep generative representations is beyond the original design of StyleGAN.

---

> ### Author Response · Authors · 2019-11-09
> **Response to Review #1 (Continued)**
>
>
> Q5: "show the results if the initial latent code w was modified directly"
> A5: Following the advice, we do experiments by (a) varying semantic from a particular level (i.e., indoor lighting) on all layers, and (b) varying semantics from different levels on layout-relevant layers. From Fig.18 of the updated submission, we can tell that when changing latent codes at all layers, other levels of semantics (such as objects inside the room) are changed. From Fig.19 of the updated submission, we can tell that when changing other semantics (e.g., category and indoor lighting) on layout-relevant latent codes, only layout but not the desired semantics varies. These two experiments further demonstrate our discovery that the early layers tend to determine the spatial layout and configuration, the middle layers control the categorical objects, and the later layers finally render the scene attributes as well as color scheme.

---

### Official Review · AnonReviewer2 · 2019-10-28
**Official Blind Review #2**

**Rating:** 3

**Review:**

The paper presents a visually-guided interpretation of activations of the convolution layers in the generator of StyleGAN on four semantic abstractions (Layout, Scene Category, Scene Attributes and Color), which are referred to as the "Variation Factors" and validates/corroborates these interpretations quantitatively using a re-scoring function. The claim of the paper is that there is a hierarchical encoding in the layers of the StyleGAN generator with respect to the aforementioned "Variation Factors".  Figure 3(a) illustrates how these "Variation Factors" emerge in the layers of the StyleGAN generator.

The basic GAN architecture used in this work is that of StyleGAN. However, details on the architecture of this particular GAN are missing, including in the Appendix. How many Convolution layers are present in its generator? Not everyone is aware of StyleGAN architecture -- A better illustration of their architecture in the main paper and its correspondence with the layer levels (bottom, lower, middle, top) is desired, mainly because the paper is built upon this. The dataset used to train the StyleGAN model is not clear either. In Appendix, Table 1 tabulates the training details, but nowhere is it clearly mentioned if the N=500,000 latent codes are sampled from a GAN model that was trained on a mixture of datasets (i.e., bedroom, living rooms, kitchen etc.) or individual datasets. As well, the unit of training time in Table 1 of the Appendix seems to be M; is it Million or Minutes? Both of them seem unrealistic units for training a GAN.

Since the training dataset is not clear, my understanding of the method is that a range of datasets are used to produce the results, especially for the effect where the transition of Semantic Category results is studied. As a first step, StyleGAN model trained on "bedroom" scenes from LSUN dataset is used to randomly sample codes from the learned distribution, which are further passed through the generator to obtain the respective image mappings. Off-the-shelf image classifiers are employed on each of the images to classify them to one of the four "Visual concepts", which is nothing but the aforementioned four semantic abstractions. Here, I would like to encourage the authors to use a consistent terminology -- The four semantic abstractions have been referred to as "Variation Factors" (page 3), "Candidate concepts" (page 6), "Visual concepts"(page 12), "Semantics" (page 8) interchangeably throughout the literature, which is confusing.  Then, 2000 top positive examples and 2000 top negative examples identified by the image classifiers are used to train a linear SVM, i.e., a binary-SVM all the four scene abstractions ("Varying Factors"), and the separation boundary is obtained. I assume the separation boundary is only obtained once, and not after every layer of the generator. Otherwise, it would not make much sense.
With the separation boundary (in the form of a normal vector) known for each of the four scene semantics, different feature activations are obtained by moving the latent code towards/away from the separation boundary. A scoring function is obtained to quantify (Equation 1) how the corresponding images vary in a particular semantic aspect when the latent code is moved from the separation boundary.  As per the last line of Paragraph 2 on page 4, a ranking of such scores using this function is used to understand the most relevant latent semantics. Does this mean that initially, a large set of semantics is used to observe whether the output of GAN is manipulated by probing each of them? Or are only four scene semantics chosen to begin with? And what happens when there is a tie? And, what value of K makes this metric more accurate? Any lower bound? Please explain. More questions on the effect of lamda later below.

In the next step, the authors sample a latent code from the learned distribution and pass it through
every layer of the GAN generator. The output code y is varied along the boundary of the SVM classifier. This is repeated at every layer of the GAN generator and the same lamda is used to perturb the resulting output code from the separation boundary. The results are visualized in Fig 3(c). The claim here is that with the same perturbation of the resulting codes (lambda=2) at the output of different GAN layers, the change in the visualized output demonstrates what kind of, if any, semantic is being captured by different layers of GAN. This is also claimed to have been validated through the "re-scoring" function. I am not very clear on this.
I request the following experiment:
1) Within just a single layer (be it bottom, lower, middle or top), how does the output change when the output code of that layer is perturbed in all directions? This is to see the effect (by visualizing) of the range of lamda values on the output at all the layers. Do you discover any changes weakening your claim?
2) I would like to see the visualizations of the latent codes at the separation boundaries, just to see how well the binary-SVM performs and whether or not, non-binary information is lost/unaccounted for.

On Page-5, see the fourth line from the bottom (going up): how do we know the desired output apriori? Are the four semantic abstractions decided based on the desired output? This takes us back to a question I asked earlier.

Moreover, Layout variation is just view-point variation. So I think it will be appropriate to call it "view-Point Variation" rather than "Layout Variation". This is because Layout is associated with spatial arrangement of objects in a scene, with functionality goals.
One last question I have is: What is so special about StyleGAN that it was used as the guiding architecture in this work? How generalizable is this approach to other kinds of GANs other than PGGAN and BigGAN (or rather, why is this approach relatable to StyleGAN, PGGAN and BigGAN alone)?

The paper has grammatical errors  (sentences are not well written), typos (ex; "manipulabe" on page-8 which should be "manipulatable") and is not polished. I also suggest the authors to change the title of the paper, which right now, is a bit odd; if you decide to keep it, there should not be a "the" before "Deep" in the title.

All in all, the paper is interesting but lacks persuasiveness.
I may jump my score if the authors address all the aforementioned questions and concerns convincingly, and work on the presentation.

**Experience Assessment:**

I have published one or two papers in this area.

**Review Assessment: Checking Correctness Of Derivations And Theory:**

I carefully checked the derivations and theory.

**Review Assessment: Checking Correctness Of Experiments:**

I carefully checked the experiments.

**Review Assessment: Thoroughness In Paper Reading:**

I read the paper thoroughly.

---

> ### Author Response · Authors · 2019-11-09
> **Response to Review #2**
>
> Thanks for the valuable comments.
>
> Q1: Details of StyleGAN.
> A1: We include the structure of StyleGAN in the appendix (Fig.17) of the updated submission. Concretely, 14 layers are used in total and "its correspondence with the layer levels (bottom, lower, middle, top)" is also discussed in Sec.E of the updated submission.
>
> Q2: "Dataset used to train the StyleGAN".
> A2: As introduced in Sec.3 "Experimental Setting" of the submission, most models are trained on a particular scene category of LSUN (e.g., bedroom model is trained only on bedroom images of LSUN). These models are used to analyze what kinds of semantics have been captured by GAN for a certain scene category, as shown in Fig.6 and Fig.7 of the submission. A mixed model is further trained on the combined set of bedroom, living room, and dining room. This model is used for category analysis, as shown in Fig.3, Fig.4 and Fig.5 of the submission. About the training time, "M" is inherited from the original paper [1], which means how many "millions" of real images seen by the discriminator. We just follow the standard. As it is not important, we remove it in the updated submission to avoid confusion.
>
> [1] A Style-Based Generator Architecture for Generative Adversarial Networks. Kerras et al., CVPR'19.
>
> Q3: "a range of datasets are used to produce the results, especially for the effect where the transition of Semantic Category results is studied".
> A3: For category transition, all images are generated from the same model, which is trained on the combined set of bedroom, living room, and dining room. We simply control the latent codes at category-relevant layers to make the image transfer from one category to another. This turns an unconditionally trained GAN into a GAN conditioned on image class, which is surprising compared to BigGAN that is designed as conditional GAN.
>
> Q4: "use a consistent terminology"
> A4: Thanks for the suggestion. We revise the submission accordingly to follow a more consistent terminology. Just to clarify, "Variation Factors", "Visual concepts", and "Semantics", all stand for human-understandable semantics, while "Candidate concepts" means the entire set of semantics, including layout prediction, 365 scene categories, 102 scene attributes, and color scheme (see Sec.A.2 of the submission). They are all employed for analysis because we don't know what kinds of semantics have been actually encoded in the latent space. The purpose of the proposed re-scoring technique is to identity the most relevant (i.e., most manipulatable) semantics from all candidates.
>
> Q5: "separation boundary is only obtained once, and not after every layer of the generator"
> A5: Separation boundary is obtained once at every layer of the generator respectively. When training the SVM boundary, the labels are obtained from the classifiers (i.e., for each factor, 2000 top positive examples are labeled as 1 and 2000 top negative samples are labeled as 0), but the training data for each layer is different. As mentioned in Sec.3.1 of the submission, StyleGAN employs different style codes for different layers. We use these style codes to train different boundaries at different layers. When manipulating images, we use the layer-specified boundaries on proper layers (e.g., if layer 2-6 are most relevant to category, when doing category transition, we simultaneously move latent code of layer 2 towards layer-2 category boundary, move latent code of layer 3 towards layer-3 category boundary, and so on and so forth).
>
> Q6: "Does this mean that initially, a large set of semantics is used to observe whether the output of GAN is manipulated by probing each of them"
> A6: Yes. See A4.
>
> Q7: "And what happens when there is a tie? And, what value of K makes this metric more accurate?"
> A7: If two semantics have the same score, we assume they are equally manipulatable. Obviously, the larger $K$ is, the more accurate the score will be. However, we care about the relative value instead of the absolute value, which means that as long as the score of one candidate is higher than another, the former one is more manipulatable than the latter. Accordingly, we just need to make sure using same $K$ and lambda for all candidates.

---

> ### Author Response · Authors · 2019-11-09
> **Response to Review #2 (Continued)**
>
>
> Q8: "This is repeated at every layer of the GAN generator and the same lambda is used to perturb the resulting output code from the separation boundary"
> A8: We would like to reaffirm that the boundary is both semantic-specified and layer-specified. Suppose we have a 14-layers StyleGAN and 100 semantic candidates, we totally predict 14 * 100 = 1400 boundaries, meaning that for each layer, we have a specific boundary for each semantic candidate. We first probe 1400 boundaries using the re-scoring technique (with same lambda), then the relative values of the scores in Eq.(1) are able to tell which concepts are most manipulatable at which layer. Based on this, when we want to manipulate a particular semantic (e.g., change scene category), we vary the latent code towards the corresponding boundary at the most relevant layers.
>
> Q9: Required Experiment 1).
> A9: Results are included in Fig.19 of the updated submission, where we can tell that when changing other semantics (e.g., category and indoor lighting) on layout-relevant codes, only layout instead of the target semantics varies. This demonstrates that lower layers only controls layout.
>
> Q10: Required Experiment 2).
> A10: Please refer to the Ablation Study in Sec.B of the submission, which reports the SVM accuracies. Experimental results turn out that almost all SVM classifiers achieve high performance, which is also the reason why they cannot be used to identify the most relevant semantics. Instead, the proposed re-scoring method can achieve this goal.
>
> Q11: "how do we know the desired output apriori?"
> A11: The four-level semantic abstractions are pre-defined according to human perception. However, which layer of GAN controls these abstractions are obtained via the proposed re-scoring technique.
>
> Q12: "Layout variation is just view-point variation"
> A12: "layout" means 3D room structure instead of renovation structure or object placements.
>
> Q13: "What is so special about StyleGAN"
> A13: To our knowledge, StyleGAN, PGGAN, BigGAN are currently the state-of-the-art deep generative models for high-resolution image synthesis. If you have other recommendations, we are glad to analyze them as well. By the way, we would like to refer a concurrent work [2] (also submitted to ICLR 2020) which analyzes BigGAN, StyleGAN, DCGAN from single level (the earliest latent space). Compared to that concurrent work, we analyze deep generative representations at multiple layers from four abstraction levels, and do experiments on StyleGAN, BIGGAN, and PGGAN, which are trained for high-resolution scene synthesis. We also propose re-scoring technique to quantitatively identify the most relevant semantics to a well-trained GAN model.
>
> [2] On the "Steerability" of Generative Adversarial Networks. In submission, ICLR'20. https://openreview.net/forum?id=HylsTT4FvB
>
> Q14: Typos.
> A14: Thanks, we fix the typos and grammatical errors in the updated submission.

---

### Official Review · AnonReviewer3 · 2019-10-31
**Official Blind Review #3**

**Rating:** 6

**Review:**

The paper proposes an approach to analyze the latent space learned by recent GAN approaches into semantically meaningful directions of variation, thus allowing for interpretable manipulation of latent space vectors and subsequent generated images.  The approach is based on using pre-trained classifiers for semantic attributes of the images at a variety of levels, including indoor room layout, objects present, illumination (indoor lightining, outdoor lighting), etc. By forming a decision boundary in the latent space for each of these classifiers, the latent code is then manipulated along the boundary normal direction, and re-scored by the classifiers to determine the extent to which the boundary is coupled to the semantic attribute.

By taking advantage of the structured composition of the latent space into per-layer contributions in the StyleGAN approach, experiments are performed to show that different levels of semantics are captured at different layers: layout being localized in lower layers, object categories in middle layers, followed by other scene attribute, and lastly the color scheme of the image in the highest layers.  A user study shows that human judgments of the coupling between layers and semantic attribute being manipulated are consistent with this observation.  A set of qualitative experiments demonstrate manipulation along several axes.  Another set of experiments demonstrate that the importance of different semantic attribute dimensions for different scene categories varies in an interpretable way, and also that certain attribute dimensions influence each other strongly (e.g. "indoor lighting" and "natural lighting"), whereas other ones are decoupled (e.g. "layout" and other dimensions).

I am somewhat positive with respect to acceptance of the paper. On the one hand, the key idea is simple, and has been demonstrated compellingly with a broad set of experiments.  On the other hand, the insight gained is fairly superficial, boiling down to the statement that the learned latent code has structure that corresponds to semantically meaningful axes of variation, and that such structure is localized to particular levels of the layer hierarchy for particular semantic axes.

There are a few small issues with the clarity of the paper that would be good to fix:
- Fig 3a: the interpretation of the vertical axis here was not clearly described in the caption or the main text
- Fig 4 caption: typo "while lindoor" -> "while indoor"
- Fig 5: the construction of the pixel area flow visualization is not explained in the caption, and needs a bit more clarity in the main text (e.g., how are multiple instances of the same class handled?)
- Fig 6: the caption could use a bit more explanation for making these plots interpretable: e.g. say what value the vertical axis is reporting
- Fig 8: same issue as above
- p8 typo: "that contacts the latent vector" -> "that concatenates the latent vector"


**Experience Assessment:**

I have read many papers in this area.

**Review Assessment: Checking Correctness Of Derivations And Theory:**

I assessed the sensibility of the derivations and theory.

**Review Assessment: Checking Correctness Of Experiments:**

I assessed the sensibility of the experiments.

**Review Assessment: Thoroughness In Paper Reading:**

I read the paper at least twice and used my best judgement in assessing the paper.

---

> ### Author Response · Authors · 2019-11-09
> **Response to Review #3**
>
> Thanks for the valuable comments.
>
> Q1: About the "fairly superficial" insight.
> A1: There are two main insights from this work. The proposed re-scoring method allows to identify the cause-effect variation factors disentangled by GANs when trained for synthesizing scenes. It reveals the synthesis mechanism of GANs and helps understand the interpretability of deep generative model. Second, we found out that GAN learns to synthesize a scene similar to how human does, i.e., first set up the layout, then add corresponding objects relevant to the scene category, such as sofa and tv to a living room, and finally render the overall style (attribute and color scheme). To the best of our knowledge, this is the first work on understanding deep generative representation from the perspective of a semantic hierarchical composition. We also conduct extensive experiments to verify this discovery.
>
> Q2: About the typos and unclear captions.
> A2: Thanks. We revise the typos and make the captions of all figures clearer in the updated submission.

---

### Decision · Program_Chairs · 2019-12-19

**Decision:**

Reject

**Comment:**

The paper proposes to study what information is encoded in different layers of StyleGAN.  The authors do so by training classifiers for different layers of latent codes and investigating whether changing the latent code changes the generated output in the expected fashion.

The paper received borderline reviews with two weak accepts and one weak reject.  Initially, the reviewers were more negative (with one reject, one weak reject, and one weak accept).  After the rebuttal, the authors addressed most of the reviewer questions/concerns.

Overall, the reviewers thought the results were interesting and appreciated the care the authors took in their investigations.  The main concern of the reviewers is that the analysis is limited to only StyleGAN.  It would be more interesting and informative if the authors applied their methodology to different GANs.  Then they can analyze whether the methodology and findings holds for other types of GANs as well. R1 notes that given the wide interest in StyleGAN-like models, the work maybe of interest to the community despite the limited investigation.  The reviewers also point out the writing can be improved to be more precise.

The AC agrees that the paper is mostly well written and well presented.  However, there are limitations in what is achieved in the paper and it would be of limited interest to the community.  The AC recommends that the authors consider improving their work, potentially broadening their investigation to other GAN architectures, and resubmit to an appropriate venue.